# Degree day models to forecast the seasonal phenology of *Drosophila suzukii* in tart cherry orchards in the Midwest U.S.

**Matthew T. Kamiyama**[1,2], **Benjamin Z. Bradford**[2], **Russell L. Groves**[2], **Christelle Guédot**[2]*

**1** Division of Applied Biosciences, Research Institute for Sustainable Humanosphere, Kyoto University, Kyoto, Japan, **2** Department of Entomology, University of Wisconsin-Madison, Madison, Wisconsin, United States of America

* guedot@wisc.edu

**Data Availability Statement:** All relevant data are within the manuscript and its Supporting Information files.

**Funding:** CG received funding for the project from the Wisconsin Department of Agriculture, Trade,

## Abstract

Spotted-wing drosophila, *Drosophila suzukii* (Matsumura) (Diptera: Drosophilidae), is an invasive economic pest of soft-skinned and stone fruit across the globe. Our study establishes both a predictive generalized linear mixed model (GLMM), and a generalized additive mixed model (GAMM) of the dynamic seasonal phenology of *D. suzukii* based on four years of adult monitoring trap data in Wisconsin tart cherry orchards collected throughout the growing season. The models incorporate year, field site, relative humidity, and degree days (DD); and relate these factors to trap catch. The GLMM estimated a coefficient of 2.21 for DD/1000, meaning for every increment of 1000 DD, trap catch increases by roughly 9 flies. The GAMM generated a curve based on a cubic regression smoothing function of DD which approximates critical DD points of first adult *D. suzukii* detection at 1276 DD, above average field populations beginning at 2019 DD, and peak activity at 3180 DD. By incorporating four years of comprehensive seasonal phenology data from the same locations, we introduce robust models capable of using DD to forecast changing adult *D. suzukii* populations in the field leading to the application of more timely and effective management strategies.

## Introduction

Spotted-wing drosophila, *Drosophila suzukii* (Matsumura) (Diptera: Drosophilidae), is an invasive economic pest of soft-skinned and stone fruit in North America, South America, and Europe [1, 2, 3, 4]. Female *D. suzukii* possess a distinctive, serrated ovipositor that allows them to infest still ripening and ripe fruit [5], unlike other species of drosophilids which can only target overripe, rotting, or damaged fruit. Along with this unique morphological feature, *D. suzukii* have high rates of reproduction, fast generation times, and quickly adapt to variable climates making them a formidable economic pest [2, 6, 7]. Damage from *D. suzukii* to susceptible fruit crops in Western U.S. states can result in annual losses up to $511 million [8]. In Minnesota, crop damage resulting from *D. suzukii* was projected at $2.2 million annually in raspberry alone [9].

and Consumer Protection Specialty Crop Block Grant #16-01. https://datcp.wi.gov/Pages/Homepage.aspx The funders had no role in study design, data collection and analysis, decision to publish, or preparation of the manuscript.

**Competing interests:** The authors have declared that no competing interests exist.

Rigorous insecticide application regimens are the most common means of *D. suzukii* management in agricultural areas of high pest pressure [10, 11]. In the Midwest, frequent insecticide treatments are implemented in *D. suzukii* affected farms and orchards during the field season with five or more applications under heavy pest pressure [12, 13]. Chemical control is a costly management option not only monetarily, but also environmentally [8, 11, 14, 15]. Growers commonly apply insecticides which specifically target adult *D. suzukii* and rotate insecticides which incorporate active ingredients that have different modes of action, which helps minimize the number of treatments per season and reduces the risk of developing resistance [12, 13]. A better understanding of this pest's seasonal phenology and the related underlying mechanisms is an instrumental step in continuously building more efficient and effective integrated pest management strategies against *D. suzukii*.

Modeling the seasonal population variations of pests from field monitoring data can help determine periods of high pest abundance [16]. Phenology models are particularly effective in describing the population dynamics of pests which maintain relatively few and synchronous generations during the year [17, 18]. *Drosophila suzukii*, however, is a highly prolific pest which has rapid rates of reproduction and several overlapping generations [7, 19]. Previously developed *D. suzukii* predictive phenology models navigated this caveat by generating developmental stage-based models [20], or physiological age-structured models approximated by degree-days (DD) [21]. Abiotic factors such as heat units (DD) and relative humidity (RH) are major drivers of *D. suzukii* development activity [7, 22, 23, 24], and incorporating these effects allows for more descriptive models. Population modeling of *D. suzukii* can provide a better understanding of this pest's population trends in a regionally specific sense, having potential management implications such as timing insecticide treatments to coincide with predictions of high pest pressure [20, 21]. Models have previously been produced from data obtained in the Pacific Coast, North Carolina, and Michigan regions [13, 20], but they do not accurately describe *D. suzukii* population dynamics observed in Wisconsin. No model incorporating DD or RH describing the pest's seasonal population variation in the Midwest is currently available.

To better understand the complex population dynamics of *D. suzukii*, we developed two complementing predictive phenology models: a generalized linear mixed model (GLMM), and a generalized additive mixed model (GAMM). Previous research involving elucidating the phenology of insects through population modelling include the use of both linear [25], and additive models [13]. These models are implemented to help observe trends in species phenology, explore the influence abiotic factors have on populations dynamics, and provide general estimates of insect population intensity [13, 16, 20, 25]. In general, GAM models are able to estimate different DD's corresponding to phenological events such as first detection or peak activity, and GLM models have the capacity to produce trap catch approximations provided a specific DD. Implementing compatible models to emphasize different aspects of *D. suzukii's* phenology furthers our understanding of how abiotic factors may driver the pest's seasonal population dynamics and allows for more precise field population trend predictions that are potentially useful for preemptive management in areas with high levels of *D. suzukii* overlapping with susceptible crops.

The GLMM and GAMM established from our study is capable of predicting the dynamic seasonal phenology of *D. suzukii* based on four years of adult monitoring trap data collected in Wisconsin tart cherry orchards during the growing season in relation to DD and RH. By incorporating four years of comprehensive seasonal phenology data from the same locations, we present two models using DD and RH which forecast changing adult *D. suzukii* populations in tart cherry orchards in Wisconsin.

## Materials and methods

### Adult seasonal phenology

**Field sites.**    To assess the seasonal phenology of *D. suzukii*, we sampled four tart cherry orchards from 2015–2018 in northeastern Wisconsin, U.S., located near the urban centers of Maplewood, Sturgeon Bay, Egg Harbor, and Sister Bay (Table 1). Sampling occurred on private land, and permission to conduct research was granted from each land owner of every site. Each year, a total of 13 traps were placed across these locations, and each trap location was resampled each year. Three traps were placed in Maplewood, five traps in Sturgeon Bay, two traps in Egg Harbor, and three traps in Sister Bay. All of the orchards grew the 'Montmorency' cultivar, while one location grew 'Montmorency' and 'Balaton'. The orchard which grew 'Balaton' was retained in the model because the 'Balaton' site had an average weekly trap catch throughout the four years (29.95 flies) that fell within the 95% confidence interval of the average weekly *D. suzukii* trap catch for all the sites throughout the four years (22.26–30.04 flies). Also, both 'Balaton' and 'Montmorency' cultivars had similar *D. suzukii* egg and larval infestation levels in lab assays from a previous study [26]. All orchards were conventionally managed for *D. suzukii* by rotating pyrethroid and organophosphate insecticides, averaging four to five applications throughout the growing season. Adult *D. suzukii* field monitoring from 2015–2018 typically began in mid-May, and continued until late-August, two weeks post tart cherry harvest.

**Monitoring traps.**    Scentry traps baited with Scentry *D. suzukii* attractant lures (Scentry Biologicals Inc., Billings, MT, U.S.) were used to monitor adult populations of *D. suzukii*. At the bottom of each trap, a drowning solution containing 200 mL of water, 0.8 g of boric acid, and 2–3 mL of unscented dish soap (Seventh Generation Inc., Burlington, VT, U.S.) was added to kill and preserve collected specimens. Monitoring traps were placed in the lower canopy fruiting zone of tart cherry trees in the interior of each orchard. Every week, the trap contents were emptied and deposited in 70% ethanol, then returned to the laboratory where adult *D. suzukii* from each trap were counted. The drowning solution was replaced weekly, and the Scentry lures were substituted every four weeks based on the manufacturer's recommendations.

**Table 1.  Orchard and weather station locations.**

| Orchard | Longitude | Latitude | Weather Station | Longitude | Latitude | Distance |
|---------|-----------|----------|-----------------|-----------|----------|----------|
| A | -87.5059 | 44.7623 | Nasewaupee | -87.5056 | 44.7597 | 0.1 km |
| A | -87.4541 | 44.7806 | Nasewaupee | -87.5056 | 44.7597 | 4.5 km |
| B | -87.4304 | 44.757 | Nasewaupee | -87.5056 | 44.7597 | 6.0 km |
| C | -87.0992 | 45.2493 | Sister Bay | -87.0662 | 45.2191 | 4.3 km |
| C | -87.0984 | 45.229 | Sister Bay | -87.0662 | 45.2191 | 2.8 km |
| C | -87.0941 | 45.2064 | Sister Bay | -87.0662 | 45.2191 | 2.6 km |
| C | -87.2698 | 45.0549 | Egg Harbor | -87.2598 | 45.0509 | 0.9 km |
| C | -87.24 | 45.0697 | Egg Harbor | -87.2598 | 45.0509 | 2.6 km |
| D | -87.3262 | 44.8773 | Sturgeon Bay | -87.3678 | 44.8935 | 3.7 km |
| D | -87.3217 | 44.8789 | Sturgeon Bay | -87.3678 | 44.8935 | 4.0 km |
| D | -87.3185 | 44.8796 | Sturgeon Bay | -87.3678 | 44.8935 | 4.2 km |
| D | -87.323 | 44.883 | Sturgeon Bay | -87.3678 | 44.8935 | 3.7 km |
| D | -87.3257 | 44.8802 | Sturgeon Bay | -87.3678 | 44.8935 | 3.6 km |

Longitude and latitude of the 13 monitoring traps from the four orchards. The weather station corresponding to each monitoring trap is also provided with its respective GPS location. 'Distance' refers to the straight-line distance between the weather station and monitoring trap.

**Temperature and humidity.** Temperature data for each site from 2015–2018 were retrieved from the PRISM Climate Group [27] by entering the GPS coordinates of each site. Daily minimum, mean, and maximum temperatures were retrieved each day from January 1st to December 31st for the years 2015–2018, and degree-days (DD) were calculated using a lower threshold of 7.2˚ C and upper threshold of 30˚ C, as *D. suzukii* development ceases beyond these temperature boundaries [7]. Cumulative weekly DD totals were computed for each site from January 1st of each year until the end of *D. suzukii* monitoring. Relative humidity (RH) data for each site from 2014–2018 were retrieved from the Michigan State Enviro-weather website [28]. Data were downloaded from four weather stations located 0.1–6.0 km from the monitoring sites (Table 1). Relative humidity was averaged for each week during the monitoring periods.

## Statistical analysis

**Generalized linear mixed model.** A generalized linear mixed model (GLMM) was gener-ated from *D. suzukii* adult trap catch, DD, RH, year, and site data. The model is based on nega-tive binomial regression fit by maximum likelihood using the *glmer* function in the *lme4* package of R version 3.5.3 [29]. Year, site, and the year*site interaction were incorporated into the model as random effects, and DD and RH were added as fixed effects. Degree days were divided by 1000 to fit an appropriate scale with the other effects. The GLMM can be described as follows:

$$Y_{ij} \sim \text{NegBin} \left[ \mu_{ij(abc)} \right]$$

$$g[\mu_{ij(abc)}] = log \left[ \mu_{ij(abc)} \right] = \mu_{ij} + \beta_i + \beta_j + \varepsilon_a + \varepsilon_b + \varepsilon_c + \delta_{ij}$$

$$\varepsilon_a \sim \text{N} \left( 0, \sigma^2_a \right),$$

$$\varepsilon_b \sim \text{N} \left( 0, \sigma^2_b \right),$$

$$\varepsilon_c \sim \text{N} \left( 0, \sigma^2_c \right),$$

$$\delta_{ij} \sim \text{N} \left( 0, \sigma^2 \right)$$

In this representation of the model, ($Y_{ij}$) is the estimate of weekly adult *D. suzukii* trap catch as a function of non-linear time (DD) ($x_i$) and RH ($x_j$) with year ($\varepsilon_a$), site ($\varepsilon_b$), and year*site ($\varepsilon_c$) added as random effects. Coefficients $\beta_i$ and $\beta_i$ correspond to the trap catch explained by DD and RH respectively, and $\delta_{ij}$ represents the residual error of the trap catch estimation. The GLMM allowed for random effects to be accounted for prior to estimating the regression coef-ficients. While separating random from mixed effects creates a much more interpretable model, this approach limits its flexibility as the coefficient estimate will be solely increasing or decreasing [30]. The GLMM is most appropriate for generating an anticipated *D. suzukii* trap count at any given week during the field season. This model as fit will only predict an increas-ing or decreasing trap catch throughout the entire seasonal phenology of *D. suzukii* based on changing DD, but can predict a specific trap catch given a unique DD estimate. The GLMM pairs well with the GAMM which provides DD approximations for critical occurrences in the field such as first adult *D. suzukii* detection, or peak activity.

**Model diagnostics.** Scatter plots were produced for trap catch at each site from 2015–2018. These scatter plots mirrored quantile-quantile plots of the same data, meaning the effect

of site and year could be assumed to be random and of equal variance [30]. The full GLM model (including RH as a fixed effect with all random effects) and the reduced model (removing RH and the interaction between random effects) were compared to determine which model best fit the data. An Akaike information criterion (AIC) and a likelihood ratio test (LRT) were run to determine the best fit model between the two models. The inclusion of all parameters was justified as the full model had slightly more explanatory power when comparing the two models (full model AIC = 4875.7, reduced model AIC = 4883.7, LRT $p$ = 0.002). A lower AIC value indicates a more parsimonious model.

**Generalized additive mixed model.** A generalized additive mixed model (GAMM) was also generated from the same *D. suzukii* adult trap catch, DD, RH, year, and site data. The model is based on negative Poisson regression with a log-link using the *predict.gam* function in the *lme4* and *mgcv* packages of R version 3.5.3. Degree days, year, site, and the year*site interaction were incorporated into the model as random effects, and RH was added as a fixed effect. The GAMM can be described as follows:

$$Y_{ij} \sim \text{Poisson} \left[ \mu_{ij(abc)} \right]$$

$$g[\mu_{ij(abc)}] = log \left[ \mu_{ij(abc)} \right] = f(d_i) + \beta_j + \varepsilon_a + \varepsilon_b + \varepsilon_c + \delta_{ij}$$

$$\varepsilon_a \sim \text{N} \left( 0, \sigma^2_a \right),$$

$$\varepsilon_b \sim \text{N} \left( 0, \sigma^2_b \right),$$

$$\varepsilon_c \sim \text{N} \left( 0, \sigma^2_c \right),$$

$$\delta_{ij} \sim \text{N} \left( 0, \sigma^2 \right)$$

In this model, ($Y_{ij}$) is used to describe the seasonal pattern of adult *D. suzukii* through non-linear time (DD/1000) ($x_i$) as a random effect and RH ($x_j$) as a fixed effect. Coefficient $\beta_i$ corresponds to the trap catch explained by RH and $f(d_i)$ is included in the model as a penalized cubic regression smoothing function of DD. Year ($\varepsilon_a$), site ($\varepsilon_b$), and year*site ($\varepsilon_c$) were also added as random effects, and $\delta_{ij}$ represents the residual error of the trap catch estimation. Using the GAMM, we have the ability to estimate underlying trends of the *D. suzukii* population throughout the growing season. Overall, the GAMM has less interpretability than the GLMM [30], but has more flexibility in attributing a specific DD to critical points in the seasonal phenology of *D. suzukii* including first adult detection, above average trap catch, and peak activity. The GAMM complements the GLMM which can predict *D. suzukii* trap catches during the field season, provided a specific DD.

**Model diagnostics.** Scatter plots were produced for trap catch at each site from 2015–2018 which mirrored quantile-quantile plots of the same data allowing us to assume the effect of site and year was random and of equal variance [30]. Similar to the GLMM, the GAM full model (including RH as a fixed effect with all random effects) and the reduced model (removing RH and the interaction between random effects) were compared to determine the best fitting model. A generalized cross-validation (GCV) score was generated for both models using the *mgcv* package of R version 3.5.3. The GCV score is used to measure model smoothness selection with respect to the smoothing parameters, as well as estimate prediction error [30]. A minimized GCV score indicates a smoother model, and in a sense, a GCV score is comparable to an AIC value in that a lower score equates to a better fitting model. The full GAM model

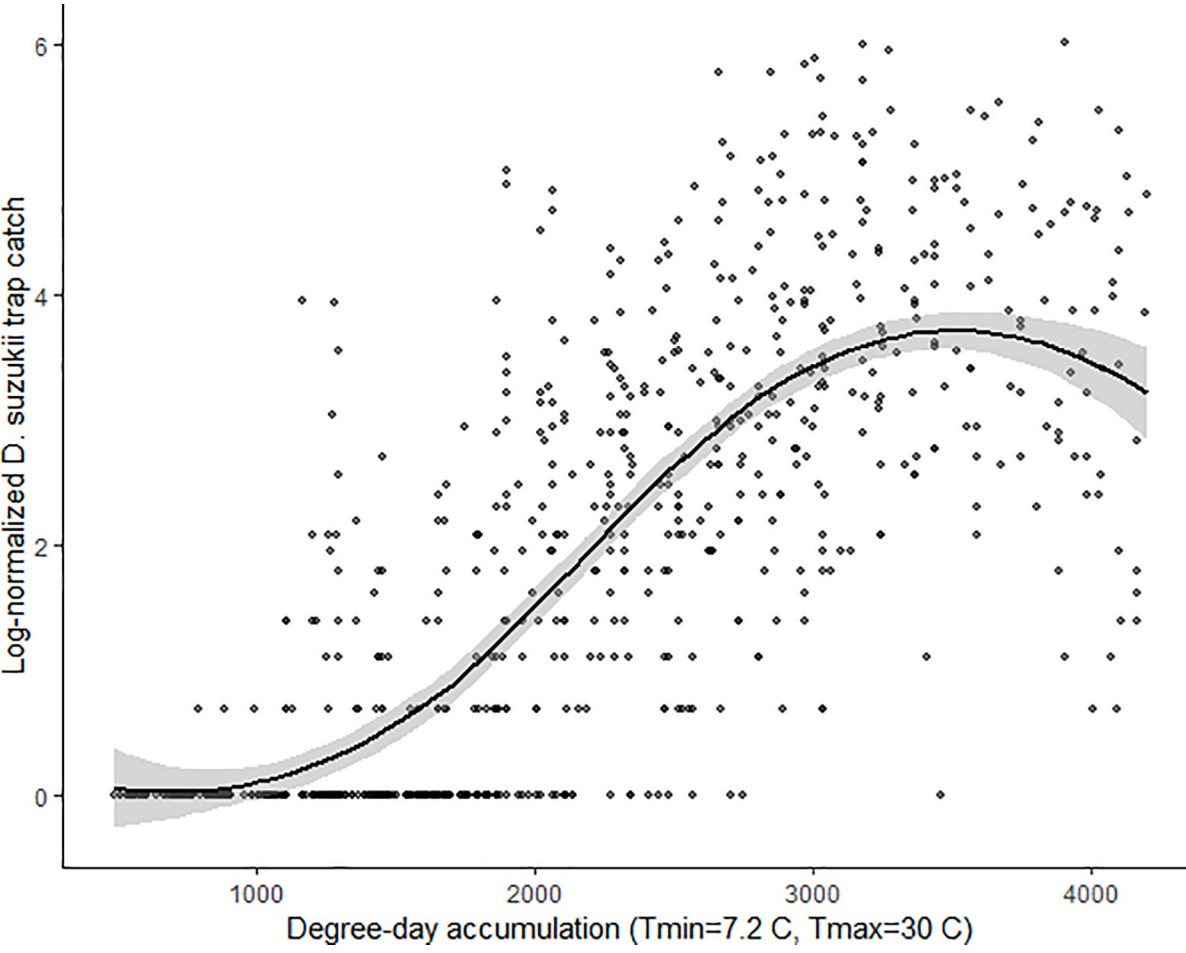

**Fig 1. Weekly adult *D. suzukii* trap catch over degree days.** Relationship between log-normalized adult *D. suzukii* total trap catch for each trap/week and weekly cumulative degree days over the four year trapping period. A smoothed fit line with 95% confidence bands illustrate the approximate phenology trend of *D. suzukii* for this dataset.

was selected for analysis as overall it had the most explanatory power (full model GSV = 1.63, reduced model GSV = 1.77).

## Results

The field populations of *D. suzukii* generally increased with DD during the field season, then peaked at the accumulation of about 3000 DD when looking at the log-transformed trap catch data from each site from 2015–2018 (Fig 1). Log-normalized average trap catch for each orchard site from the four year study period is presented with best fit curves to better illustrate the trap catch variance between each orchard (S1 Fig). A simple linear regression explained that the log-normalized trap catch had a weak positive correlation with RH ($t = 9.23$, $p < 0.001$, $R^2 = 0.10$, $y = 0.12x - 7.16$) (Fig 2).

The GLMM analyzed trap catch from each site over the four year period. The random effect of year explained 24% of the total variance in *D. suzukii* trap catch, while the effect of site accounted for 5%, and the year * site interaction was responsible for 18% of the trap catch variability of the random effects. The coefficient estimates are 2.21 ($Z = 23.95$, $p < 0.001$) for DD/1000 and 0.03 ($Z = 0.02$, $p = 0.08$) for RH. This model estimates an increase of $e^{(2.21)}$ or about

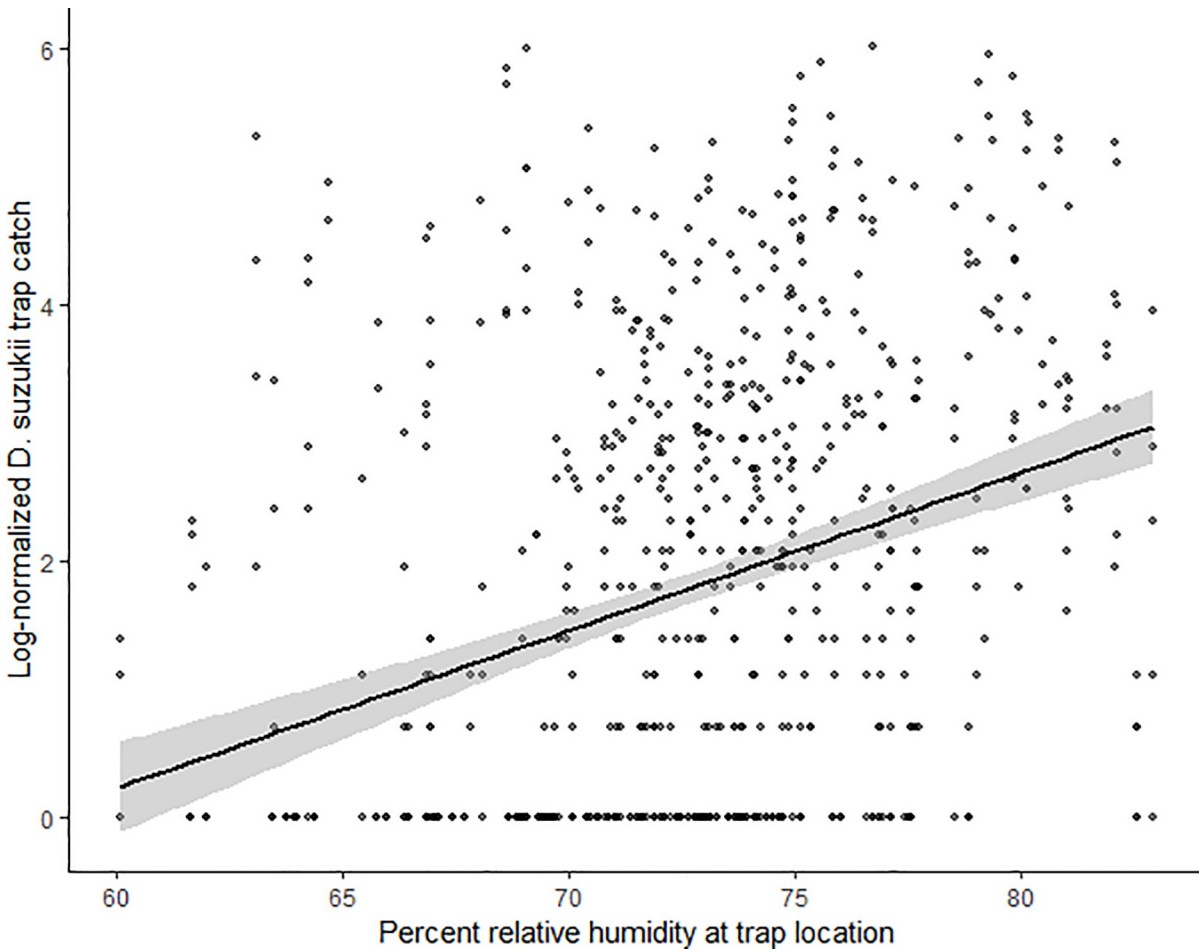

**Fig 2. Weekly adult *D. suzukii* trap catch over relative humidity.** Relationship between log-normalized total adult *D. suzukii* trap catch for each trap/week and mean weekly relative humidity at the trap location over the four year trapping period. A weak positive relationship between trap catch and increasing relative humidity is shown by the smoothed fit line with 95% confidence bands.

nine flies per every 1000 DD accumulated and an increase of $e^{(0.03)}$ or about one fly per every increase of one percent RH.

A smoothed curve plot (Fig 3) was generated by the GAMM, incorporating trap catch data from all sites over the four years to analyze seasonal population trends of *D. suzukii*. The Y-axis is represented by conditional modes (CM), which measure trends at the population level given the effects provided in the model. The plot suggests first adult detection occurs at 1276 DD and periods of above average trap catch began at 2019 DD and continued until 4707 DD. A very distinguishable interval of *D. suzukii* population increase occurs between 1549 DD and 3180 DD, which roughly corresponds with early July through early September in Wisconsin. This population increase is denoted by the increasing CM from negative (below average) to positive (above average) between 1549 DD and 3180 DD.

The GAMM produced critical DD values (1276, 1549, 2019, and 3180) were plugged into the GLMM with their corresponding RH values and a predicted trap catch was generated (Table 2). Relative humidity values were taken from the sampling event closest in DD to the critical DD values from each site and year and were averaged. The true *D. suzukii* trap catches were averaged from each site and year from the sampling event closest to each critical DD value (Table 2). One sample *t*-tests were then performed to determine the validity of the

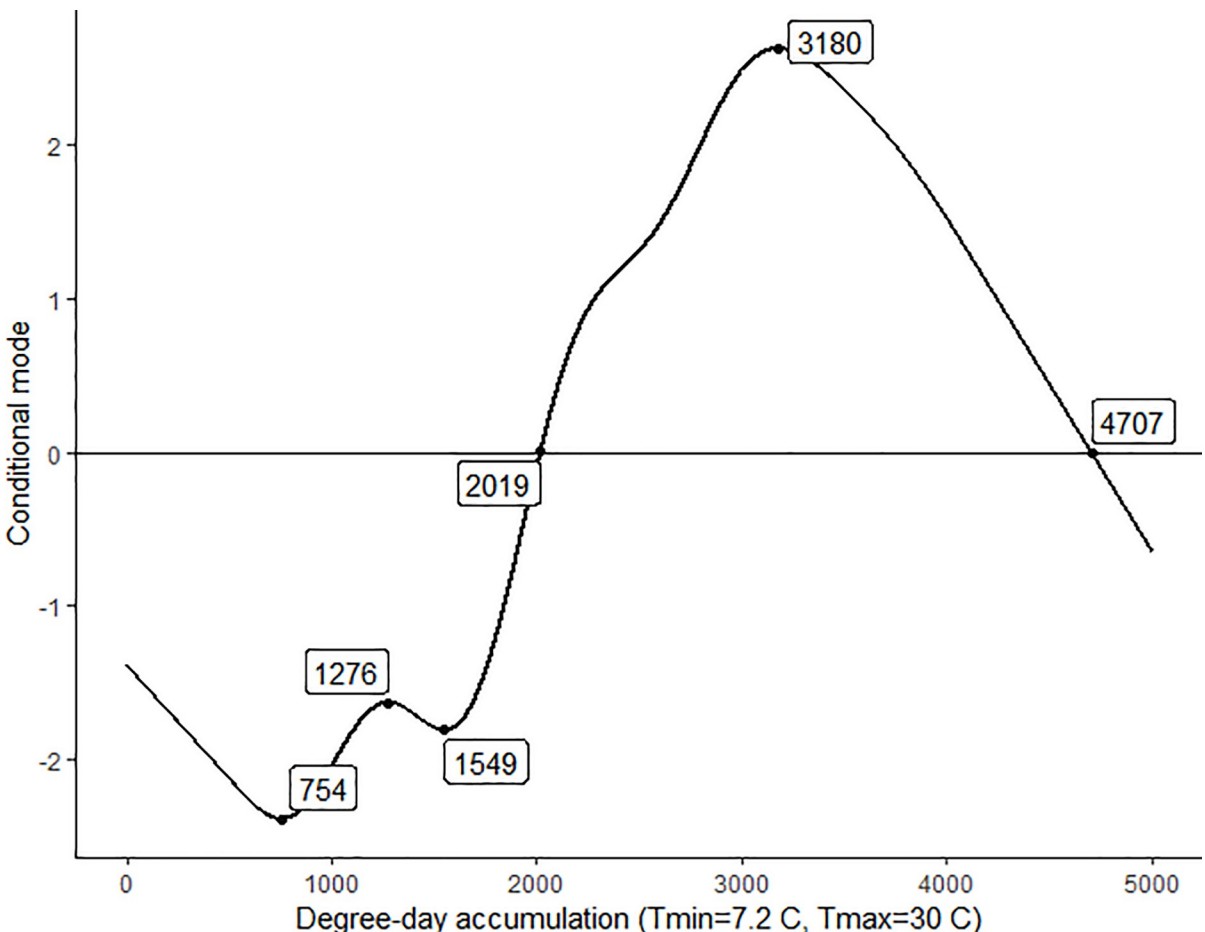

**Fig 3. GAMM predicted *D. suzukii* population dynamics over degree days.** GAMM generated smooth curve plot of adult *D. suzukii* trap catch total for each site in relation to degree day accumulation (DD) over the four year trapping period. Critical DD values are labelled and boxed on the curve. Conditional modes (CM) measure the population level estimations given the effects (positive CM = higher than average, negative CM = lower than average).

GLMM by comparing the predicted mean trap catches to the true mean trap catches. The GLMM predicted trap catches at 1276 and 3180 DD were statistically similar to the corresponding true trap catches at $p < 0.05$, but the predicted trap catches at 1549 and 2019 DD were significantly different than the true trap catches at those respective DD at $p < 0.05$.

**Table 2. Comparisons of GLMM predicted adult *D. suzukii* trap catch with true trap catch.**

| DD | RH | GLMM trap catch | True trap catch | 95% CI | t | df | p |
|---|---|---|---|---|---|---|---|
| 1276 | 70.7 | 1.2* | 3.1 ± 1.2 | 0.7–5.6 | 1.6 | 51 | 0.1 |
| 1549 | 71.9 | 2.5 | 1.2 ± 0.4 | 0.4–1.9 | -3.7 | 51 | 0.001 |
| 2019 | 71.7 | 6.6 | 13.7 ± 3.5 | 6.6–20.7 | 2.0 | 51 | 0.05 |
| 3180 | 75.4 | 87.1* | 95.9 ± 16.2 | 63.2–128.6 | 0.5 | 41 | 0.6 |

*Drosophila suzukii* mean trap catch estimation resulting from the GLMM (generalized linear mixed model) compared to the actual mean trap catch (± SE) from all sites from 2014–2018 at each of the critical DD (degree day) value, and corresponding average RH (relative humidity). Results from the one sample *t*-test are also included: *t*-test statistic (*t*), degrees of freedom (*df*), *p*-value (*p*), and 95% confidence interval (CI) of the true mean trap catch.

\* indicates a GLMM predicted *D. suzukii* trap catch statistically similar to the true trap catch at the corresponding DD and RH (one sample *t*-test: $p < 0.05$).

## Discussion

The models presented from this research suggest that seasonal *D. suzukii* populations in Wisconsin tart cherry orchards follow predictable patterns based on DD and RH. Our models integrate a robust dataset comprising four years of *D. suzukii* trap catches from the same tart cherry orchards in eastern Wisconsin. The GLMM model is able to forecast adult *D. suzukii* numbers as weekly trap catches, which is useful when needing quantitative estimates of field populations. The GAMM model generated a smoothed curve plot representing *D. suzukii* population tendencies throughout the growing season, which is valuable in determining periods of high or low risk for elevated *D. suzukii* numbers and phases of rapid population increase or decrease. The field population trends described from this work are characteristic of *D. suzukii* phenology in the Midwest, as studies in Wisconsin and Michigan fruit crops report first adult detection in the early summer, peak activity in the late summer, and decreasing populations in the early fall [13, 23, 26, 31]. It is worth noting that in our study monitoring typically ceased a few weeks after tart cherry harvest, limiting our ability to extrapolate on the population trends beyond early fall and into winter. Our seasonal population trend findings differ from previous *D. suzukii* phenology work done in California which describe a bimodal population distribution with peak numbers occurring in the early summer and late fall, and a decrease in population during the mid to late summer [32]. It is possible this mid-season period of quiescence in California *D. suzukii* populations is explained by the flies' exposure to high temperatures and low humidity; similar conditions are suggested in inducing a vernal-estival dormancy in other Diptera, such as the olive fruit fly (*Bactrocera oleae*) [33].

The findings of our research are comparable to *D. suzukii* population modeling studies from Michigan, Washington, Oregon, and North Carolina [13, 20]. In the work done by Wiman et al., 2014, estimates for California and North Carolina populations of *D. suzukii* followed a bimodal trend of early summer increase, mid-summer decrease, then late summer increase, whereas the Oregon *D. suzukii* estimates were more analogous to our results of a single peak of high activity late in the field season. However, the data used to predict *D. suzukii* populations in Oregon were largely different than our Wisconsin data; the highest *D. suzukii* trap catch was 200 adults per week in Wisconsin compared to 27 in Oregon, and around 4000 DD were accumulated in Wisconsin during the field season while roughly 1900 DD (Tmin = 4° C, no Tmax) were accumulated in Oregon [20]. A predictive generalized additive model (GAM) generated from seven years of *D. suzukii* trap catch data in Michigan blueberry forecasted a similar population trend to our findings of first *D. suzukii* detection occurring in the early/mid-summer, populations peaking in the late summer, then numbers decreasing in the late summer/early fall [13]. The GAM developed by Leach et al., 2019 was based off of calendar day, as opposed to our GAMM based off DD, and incorporated parameters including: first *D. suzukii* capture, spring activity, prior year max activity, the number of days below 0°C and above 10°C in spring and winter, the number of days above 60% RH, and the number of days above 28°C. According to their GAM, first *D. suzukii* catch was heavily influenced by the number of days below 0°C and above 10°C, and peak capture was strongly influenced by all measured parameters [13]. Our GAMM estimated first detection to occur at 1262 DD (mid to late June) similar to the GAM predictions in Leach et al., 2019. Our GAMM predicted peak activity at 3180 DD (early September) contrasting from that of Leach et al., 2019, which predicted peak activity in late September/early October [13]. The earlier peak activity forecasted by our model is potentially an artifact of tart cherries being an earlier ripening fruit crop than blueberries [13], demonstrating the importance of host crop availability in *D. suzukii* phenology.

The predictive models presented from our research are supported by regional and crop specific data obtained from Wisconsin tart cherry orchards, and are the first which forecast field

*D. suzukii* population trends in the Midwestern U.S. relative to DD that could be used for pre-emptive management strategies. Our DD based models allow for a more precise estimation of field *D. suzukii* populations than calendar day models alone, but incorporating DD may restrict the ability to discern between the different late season generations of this multivoltine pest [7, 20]. One advantage of structuring models with DD is that they do not assume consistent temperatures from year to year on the same date, and can be more easily applied to other locations with dissimilar climates. Since the life cycle of *D. suzukii* is so heavily dependent on temperature [7, 23, 34], we assumed there would be less variability in trap catch from year to year for a given DD as opposed to calendar day. Relative humidity data from each site each year were also incorporated into the models as RH also plays an important role in *D. suzukii's* seasonal activity [22, 23, 24]. Several additional factors such as photoperiod, host crop availability, and presence of natural enemies may impact the phenology of *D. suzukii* as well [21, 35, 36]. Another factor which may affect *D. suzukii* field population dynamics that is commonly overlooked is the frequency of insecticide applications. Insecticide applications are typically initiated with first *D. suzukii* detection in the early summer when fruit is beginning to ripen, and continue until the completion of harvest [10]. Our study, along with previous work done on *D. suzukii* phenology in the Midwest [13, 23, 26, 31], acknowledge, but did not incorporate pesticide use as an effect when analyzing *D. suzukii* population dynamics, meaning the influence insecticide application has on trap catch in this region is largely unknown. Future models directed towards approximating the seasonal phenology of *D. suzukii* should consider all significant biotic and abiotic effects, including factors that remain unresolved such as *D. suzukii* overwintering habits, impact of native biological control, and host crop characteristics.

The GLMM is slightly more accurate in estimating *D. suzukii* populations in the early and later portions of the season (Table 2), meaning first detection and peak activity can be more precisely predicted than mid-season numbers in affected Midwest tart cherry orchards. This may be due to the fact that the GLMM will only estimate an increasing or decreasing population trend. As a result, the model was unable to acknowledge the brief decrease in trap catch (1276–1549 DD) after first *D. suzukii* detection and before rapid population increase.

According to the GAMM, the first detection of *D. suzukii* is estimated to occur at 1262 DD, meaning control measures should be initiated when that DD accumulation is reached and fruit has reached a susceptible stage in a given year [26]. For reference, 1276 DD fell most closely on June 26[th], 24[th], 23[rd], and 22[nd] in 2015, 2016, 2017, and 2018 respectively. Field larval infestations of tart cherries in Wisconsin have been documented starting in mid-July, roughly one month following first adult *D. suzukii* detection [26]. There also exists a specific period in the mid-summer (1549 DD) in which *D. suzukii* populations begin to increase, then finally peak in the early fall (3180 DD). Extensive management strategies implemented during the transitory 1276–1549 DD period when *D. suzukii* levels are still low may help to alleviate the impending rapid increase of *D. suzukii* in the field forecasted by our models. Wiman et al., 2016 also alludes to the importance of targeting the initial *D. suzukii* adults for control to reduce the opportunity for high populations to build up and fruit damage to increase throughout the season. Though, any management practice directed to a given crop at this time should be imposed only when susceptible fruit is present on the crop, otherwise the attempt may be meaningless.

This research adds a valuable new tool to crop protection against *D. suzukii* by predicting their field population trends in relation to DD an RH, allowing for the preemptive implementation of integrated pest management strategies, or improved trap deployment. Our models best apply to growers in the Midwest who farm tart cherries in locations with a known presence of the pest, or areas of potential invasion with a comparable climate to the Midwest U.S. Currently, growers in the Midwest are advised to begin management practices at first detection

of *D. suzukii* in their region [37], but in some cases, control measures in some fruit crops may be delayed until fruits reach a susceptible stage of development [26]. We understand that these models are regionally limited, derived from the specific, and varying factors pertaining to Wisconsin tart cherry orchards. However, continued work on population modeling of *D. suzukii* in different climatic areas with the inclusion of crop susceptibility and fruit field infestations is an integral step towards the effective and efficient management of this pest in all invaded regions.

## Supporting information

**S1 Fig. Weekly adult *D. suzukii* trap catch separated by site over degree days.** Relationship between log-normalized adult *D. suzukii* total trap catch for each orchard site/week and weekly cumulative degree days over the four year trapping period. A smoothed fit line shows approximate phenology of *D. suzukii* for each orchard in this dataset.
(TIFF)

**S1 Data.**
(XLSX)

**S2 Data.**
(XLSX)

**S3 Data.**
(XLSX)

**S4 Data.**
(XLSX)

## Acknowledgments

We thank all participating growers for access to their orchards to set up *D. suzukii* monitoring traps. We also thank Matt Stasiak from the University of Wisconsin Door County Peninsular Agricultural Research Station for helping with trapping efforts and providing us with archived trap catch data. We give special thanks to Zach Schreiner, Janet van Zoeren, and Benjamin Jaffe for their help with data processing and field work.

## Author Contributions

**Conceptualization:** Russell L. Groves, Christelle Guédot.

**Data curation:** Matthew T. Kamiyama.

**Formal analysis:** Matthew T. Kamiyama, Benjamin Z. Bradford.

**Funding acquisition:** Christelle Guédot.

**Methodology:** Matthew T. Kamiyama, Benjamin Z. Bradford, Russell L. Groves, Christelle Guédot.

**Project administration:** Christelle Guédot.

**Writing – original draft:** Matthew T. Kamiyama.

**Writing – review & editing:** Matthew T. Kamiyama, Benjamin Z. Bradford, Russell L. Groves, Christelle Guédot.

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
