## [Decision Letter · Decision Letter 0]

12 Feb 2020

PONE-D-19-35638

Degree day models to forecast the seasonal phenology of Drosophila suzukii (Diptera: Drosophilidae) in Midwest climate

PLOS ONE

Dear Dr. Guédot,

Thank you for submitting your manuscript to PLOS ONE. After careful consideration, we feel that it has merit but does not fully meet PLOS ONE’s publication criteria as it currently stands. Therefore, we invite you to submit a revised version of the manuscript that addresses the points raised during the review process.

Your manuscript has been reviewed by three independent and qualified reviewers. They returned with contrasting recommendations (minor, major revisions and reject). Some comments are apparently serious, e.g., trap position, unconsidered factors (variety, pesticide applications, etc). Therefore, I need to read a full point by point rebuttal letter to the comments below, before taking any further editorial decision.

Specific comments from the academic editor:

- I suggest adding the crop system in the title. Moreover, although Midwest is pretty known I suggest adding US. (Diptera: Drosophilidae) could be deleted.

- the raw data file gives errors in most cells with formulae

We would appreciate receiving your revised manuscript by Mar 28 2020 11:59PM. To enhance the reproducibility of your results, we recommend that if applicable you deposit your laboratory protocols in protocols.io, where a protocol can be assigned its own identifier (DOI) such that it can be cited independently in the future. For instructions see: http://journals.plos.org/plosone/s/submission-guidelines#loc-laboratory-protocols

We look forward to receiving your revised manuscript.

Kind regards,

Antonio Biondi, Ph.D.

Academic Editor

PLOS ONE

Journal Requirements:

Additional Editor Comments (if provided):

Your manuscript has been reviewed by three independent and qualified reviewers. They returned with contrasting recommendations (minor, major revisions and reject). Some comments are apparently serious, e.g., trap position, unconsidered factors (variety, pesticide applications, etc). Therefore, I need to read a full point by point rebuttal letter to the comments below, before taking any further editorial decision.

Specific comments from the academic editor:

- I suggest adding the crop system in the title

- the raw data file gives errors in most cells with formulae

- please check for any reviewers' attachment

Reviewers' comments:

Reviewer's Responses to Questions

**Comments to the Author**

1. Is the manuscript technically sound, and do the data support the conclusions?

Reviewer #1: Partly

Reviewer #2: Partly

Reviewer #3: Partly

2. Has the statistical analysis been performed appropriately and rigorously? 

Reviewer #1: N/A

Reviewer #2: Yes

Reviewer #3: No

3. Have the authors made all data underlying the findings in their manuscript fully available?

Reviewer #1: Yes

Reviewer #2: Yes

Reviewer #3: No

4. Is the manuscript presented in an intelligible fashion and written in standard English?

Reviewer #1: No

Reviewer #2: Yes

Reviewer #3: Yes

5. Review Comments to the Author

Reviewer #1: The manuscript “Degree day models to forecast the seasonal phenology of Drosophila suzukii (Diptera: Drosophilidae) in Midwest climate” analyzes two predictive models (GLMM and GAMM) on the dynamic seasonal phenology of spotted-wing drosophila (SWD) based on four years of data. Furthermore, the authors of the manuscript have a useful amount of data about the SWD captures through the season to consider the fly abundance during the season. However, the manuscript is a bit confusing, and it is hard, at least for this reviewer, to follow the hypotheses, methodology and results obtained. In this sense, the manuscript would improve if it had a more precise distribution and data analyzed. Otherwise, it is complicated to evaluate the findings. My comments below try to explain the significant part of the shortcuts that I found during the reading.

Introduction:

The introduction needs more information about why the authors used two Generalized different models for the predictions.

Methods:

This section needs more detail about field information (maybe a figure, should help to do some visual idea about the field distribution).

There are 13 different locations with the cherry variety ‘Montmorency’; however, there one location (authors didn’t said which site is) with two different cherry varieties ‘Montmorency’ and ‘Balaton’. For this reason, this location should be excluded from the predictive models. The populations' capture in this field could increase or decrease (attract/repel SWD) in correlation with variety absent in another experimental field.

The abiotic conditions are essential if this study is focused on predictive models that consider temperature and relative humidity as a key factors. However, the authors recorded these data from weather stations “located 1-7 km” from the monitoring sites, which is of low accuracy to make these models reliable.

In general, both models are not well described. Furthermore, the model diagnostics section is confusing.

Results

L 178: “slight positive correlation with RH”; At least for this reviewer, a correlation value of R2= 0.04, is almost no correlation. However, authors in the figure text said: “A positive relationship between trap catch and increasing relative humidity is shown”, which is a bit confusing for the reader.

The mean of SWD trapped predicted in the GLMM (Table 1) does not fit well with the “true trap catch”, which is worrying. At 1549 DD, the GLM model predicts that there are 6.6 D. suzukii adults/trap but in reality, the data showed a twice flies captures (13.7 D. suzukii adults/trap). This data could make the GLMM not a consistent model to follow the SWD dynamics in cherry.

Discussion

Finally, other factors influencing SWD dynamics were mostly not detailed or discussed and final conclusions regarding how predictive models could support crop protection against D. suzukii were premature given the data presented.

Reviewer #2: PONE-D-19-35638

I appreciated the approach of this paper. It is well-written, though I had a few copy edit comments. The main issue is lack of independent, validating data and potential over-reach of the conclusions. The data are limited to a very specific crop system and the results should not be generalized to the landscape level. One of the major oversights is the lack of discussion about the potential impact of the insecticide applications on the number of adult flies trapped, even though the introduction stated that there could be five or more applications. Tart cherries often use rather strong chemistries such as dimethoate that could have a huge influence on the number of flies trapped. Also I would have liked to know if traps were on orchard borders or in interiors, as the spatial distribution of traps is known to influence captures. Finally, it would have been great to have some observations linked to the trap capture data. Most growers are giving up on traps, they are too cumbersome and time consuming. The most important variable is the infestation risk to the crop, not how many are caught in traps so that was perhaps a missed opportunity. It would be great to see if the model could predict first infestation, or total infestation on managed and/or untreated crop. That said, the data and analysis presented are relatively sound and meet the criteria for publication in PLOS. I recommend acceptance after minor changes are implemented. I had two uploaded documents: 1) marked up manuscript, and 2) line number comments corresponding to the marks on 1.

• Highlight, page 9

L28 prominent - common?

• Highlight, page 9

L31 with five or more applications under heavy pressure

• Highlight, page 9

L35 risk of developing resistance

• Highlight, page 10

L46 heat units

• Highlight, page 10

L50 to coincide with predictions of

• Strike Out, page 10

L53 the

• Highlight, page 10

L54 any reasons why phenology might be different in the midwestern US?

• Highlight, page 11

L69 would be nice to report GPS points for sites, latitude info could prove useful to others

• Highlight, page 11

L88 What resolution and time scale were the PRISM data?

• Highlight, page 11

L90 degree-days (DD) - what method used for DD?

• Highlight, page 13

L33 but wouldn't this be more in the SWD foraging environement rather than ambient? Were the orchards irrigated?

• Strike Out, page 14

L154 we?

• Highlight, page 16

L183 explanation for the confidence interval?

• Highlight, page 16

L186 explanation for CI missing

• Highlight, page 17

L206 smoothing protocol? Loess?

• Highlight, page 18

L244 but presumably if there was a later crop or host, the populations would likely increase well beyong the monitored period (Oct), or at least you don;t know from this data set, these data are specific to the tart cherry crop and should not be expanded to generalize the whole population

• Highlight, page 21

L297 Careful, no data on crop infestation are presented. What if damage preceded trap captures? Unhappy growers.

• Highlight, page 21

L309 Would be important to link the model to 1) crop susceptibility (color), 2) infestation/damage.

Reviewer #3: Degree day models to forecast the seasonal phenology of Drosophila suzukii

(Diptera: Drosophilidae) in Midwest climate (Kamiyama et al.)

The paper seeks to develop a general model for SWD in the Midwest by

fitting GLMM and GAMM statistical models to four years of field trap catch

data using degree days, RH, year, and site as independent variables. The

paper is well written and the statistical analyses well done.

The approach of putting field data on a dd scale is old (e.g., R.D. Hughes

and N. Gilbert @1968), and normalizing the data for each location and year

helps to put the phenology on a common scale based on the temperatures

experienced at the different sites and years (L 274-276). The first question

is what is the starting time for accumulating dd, and further how were the

dd calculated (the term non-linear time gives direction but not precise

information). Statistical models tend to be time and place specific

explaining why similar models from other locations do no work for the

Midwest (lines 252-265). In fact, the proposed model(s) may not work in

other areas of the Midwest. The parameter values should be given for the

models.

Mechanistic models have been developed that use biodemographic functions to

estimate the effects of physical factors (temperature, RH, photoperiod,

etc.) at short time steps on the dynamics of the target poikilotherm. In an

age -stage context, these models use abiotic and biotic factors as drivers

of the dynamics (doi:10.1016/j.ecolmodel.2016.05.014 and

doi:10.1007/s10530-016-1255-6). No model should be used to forecast

densities, hence the comment L 268-271 is inappropriate – at best one can

attempt to predict phenology “ for preemptive – IPM”. The authors recognize

this limitation L-308-309.

Some specific points:

L178- (t = 5.50, p < 0.001, are high relative to R2 = 0.04 – please check.

L201-203 This result is the critical result, and merits deeper explanation

of the conditional mode.

The work seeks to develop IPM relevant information, but fail to provide

accessible rules. Figures 1 and 2 were not particularly useful, showing a

very wide scatter of data interpreted statistically in the text.

The data reported in the supplemental materials is for 2018.

*** Note from Editorial Office: please note that the responses to the following questions are 'partly' but this is not an option in the reviewer form and 'No' has therefore been selected:

2. Has the statistical analysis been performed appropriately and rigorously? Partly

3. Have the authors made all data underlying the findings in their manuscript fully available? Partly -2018 ***

6. PLOS authors have the option to publish the peer review history of their article (what does this mean?). If published, this will include your full peer review and any attached files.

Reviewer #1: No

Reviewer #2: No

Reviewer #3: No

---

## [Author Response · Author response to Decision Letter 0]

20 Mar 2020

Authors Response to Review Comments

PONE-D-19-35638

Degree day models to forecast the seasonal phenology of Drosophila suzukii (Diptera: Drosophilidae) in Midwest climate

PLOS ONE

Dear Dr Biondi, 

We are pleased to submit for publication the revised version of “Degree day models to forecast the seasonal phenology of Drosophila suzukii (Diptera: Drosophilidae) in Midwest climate”. We appreciate all the thoughtful and constructive criticisms from the reviewers. We have addressed each of their concerns as detailed below as much as possible. The reviewers’ comments were italicized with our response directly below. We believe that the revised version can meet the criteria for publication in PLOS ONE.

Specific comments from the academic editor:

- I suggest adding the crop system in the title. Moreover, although Midwest is pretty known I suggest adding US. (Diptera: Drosophilidae) could be deleted.

We agreed with the suggestions and added the crop system and country to the title. “(Diptera: Drosophilidae)” was deleted.

- the raw data file gives errors in most cells with formulae

This was fixed.

Could you therefore please include the title page into the beginning of your manuscript file itself, listing all authors and affiliations?

A title page was included within the main text file.

Please check for any reviewers' attachment

Attachments were reviewed.

Response to Reviewer 1’s comments

The manuscript “Degree day models to forecast the seasonal phenology of Drosophila suzukii (Diptera: Drosophilidae) in Midwest climate” analyzes two predictive models (GLMM and GAMM) on the dynamic seasonal phenology of spotted-wing drosophila (SWD) based on four years of data. Furthermore, the authors of the manuscript have a useful amount of data about the SWD captures through the season to consider the fly abundance during the season. However, the manuscript is a bit confusing, and it is hard, at least for this reviewer, to follow the hypotheses, methodology and results obtained. In this sense, the manuscript would improve if it had a more precise distribution and data analyzed. Otherwise, it is complicated to evaluate the findings. My comments below try to explain the significant part of the shortcuts that I found during the reading.

We appreciate the review and comments and will address the review based on the specific comments below. 

Introduction:

The introduction needs more information about why the authors used two Generalized different models for the predictions.

More detailed reasoning behind implementing the two different models provided L74 – 87.

Methods:

This section needs more detail about field information (maybe a figure, should help to do some visual idea about the field distribution).

There are 13 different locations with the cherry variety ‘Montmorency’; however, there one location (authors didn’t said which site is) with two different cherry varieties ‘Montmorency’ and ‘Balaton’. For this reason, this location should be excluded from the predictive models. The populations' capture in this field could increase or decrease (attract/repel SWD) in correlation with variety absent in another experimental field.

Table added (Table 1) for field distribution and trap set up clarification. Justification added for retaining the ‘Balaton’ site in the models L104 – 109. The weather station locations and distance to the specific traps were added to the table for clarity.

The abiotic conditions are essential if this study is focused on predictive models that consider temperature and relative humidity as a key factors. However, the authors recorded these data from weather stations “located 1-7 km” from the monitoring sites, which is of low accuracy to make these models reliable.

We realize that the distance from the weather stations to the traps may be a concern, however, the selected weather stations were the closest relative humidity measuring devices to the trapping sites as there were no individual relative humidity tracking devices placed in the field near each trap. The sites are large orchards with weather stations on site or near the sites and using data from the weather stations for our models was our most accurate option. The furthest distance between weather station and trap was 6.0 km, all the other weather stations were between 0.1 and 4.5 km from their respective traps. The temperature data for each site was retrieved from the PRISM Climate Group by entering the GPS coordinates of each site. We added Table 1 to display each site and weather station. 

In general, both models are not well described. Furthermore, the model diagnostics section is confusing.

We added some more detail to the model descriptions and diagnostics to help with this comment. The other two reviewers did bring this up so we feel that the descriptions are appropriate as edited.

Results

L 178: “slight positive correlation with RH”; At least for this reviewer, a correlation value of R2= 0.04, is almost no correlation. However, authors in the figure text said: “A positive relationship between trap catch and increasing relative humidity is shown”, which is a bit confusing for the reader.

The mean of SWD trapped predicted in the GLMM (Table 1) does not fit well with the “true trap catch”, which is worrying. At 1549 DD, the GLM model predicts that there are 6.6 D. suzukii adults/trap but in reality, the data showed a twice flies captures (13.7 D. suzukii adults/trap). This data could make the GLMM not a consistent model to follow the SWD dynamics in cherry.

We thank you for pointing this out. The linear regression was performed on ‘trap catch’ and ‘RH’ as opposed to ‘log-normalized trap catch’ and ‘RH’. The results of the linear regression have been updated and the new R2 (R2=0.1) value has been provided L227-228.

At 1549 DD, the GLMM predicts 2.5 D. suzukii/trap. This is likely because our data show a slight decrease in trap catch from 1276 DD to 1549 DD (3.1 to 1.2 flies). However, the GLMM predicts a strictly increasing trap catch throughout the season so the GLMM was unable to predict the slight drop in flies. This is a potential oversight in the model that we added to the Discussion (L360 – 363). At 2019 DD, the GLMM predicts 6.6 D. suzukii/trap, which is on the cusp of the 95% CI of the true trap catch (6.6 – 20.7 flies). Figure 3 was also updated so the critical points in the figure now reflect those reported in the text.

Discussion

Finally, other factors influencing SWD dynamics were mostly not detailed or discussed and final conclusions regarding how predictive models could support crop protection against D. suzukii were premature given the data presented.

More details about alternative factors influencing D. suzukii dynamics added to the Discussion (L344 – 357), and final conclusions were tempered back.

Response to Reviewer #2’s comments

I appreciated the approach of this paper. It is well-written, though I had a few copy edit comments. The main issue is lack of independent, validating data and potential over-reach of the conclusions. The data are limited to a very specific crop system and the results should not be generalized to the landscape level. One of the major oversights is the lack of discussion about the potential impact of the insecticide applications on the number of adult flies trapped, even though the introduction stated that there could be five or more applications. Tart cherries often use rather strong chemistries such as dimethoate that could have a huge influence on the number of flies trapped. Also I would have liked to know if traps were on orchard borders or in interiors, as the spatial distribution of traps is known to influence captures. Finally, it would have been great to have some observations linked to the trap capture data. Most growers are giving up on traps, they are too cumbersome and time consuming. The most important variable is the infestation risk to the crop, not how many are caught in traps so that was perhaps a missed opportunity. It would be great to see if the model could predict first infestation, or total infestation on managed and/or untreated crop. That said, the data and analysis presented are relatively sound and meet the criteria for publication in PLOS. I recommend acceptance after minor changes are implemented. I had two uploaded documents: 1) marked up manuscript, and 2) line number comments corresponding to the marks on 1.

Thank you for the comments. The conclusions made in the Discussion have been clarified to apply to specifically tart cherry crop systems. Potential impact of insecticides was acknowledged (L351 – 354). Added ‘host crop characteristics’ to list of unresolved effects that future models should consider to account for tart cherry chemistry. Trap location was added to Methods. We agree, it would have been informative to measure fruit infestation along with trap catch throughout the season. However, it is difficult to collect tart cherries when they are in a susceptible stage of development. In our past work with tart cherries and D. suzukii field larval infestations, we only managed two collections which yielded larval infestations (collecting fruit every other week). Growers want to minimize the amount of time ripe fruits are exposed in the field so they are harvested quickly reducing the time to sample. In the future, multiple trees should be left un-harvested for fruit sampling throughout the season for every year of the phenology study to gain a better understanding of field larval infestations.

• Highlight, page 9

L28 prominent - common?

Changed.

• Highlight, page 9

L31 with five or more applications under heavy pressure

Changed.

• Highlight, page 9

L35 risk of developing resistance

Fixed.

• Highlight, page 10

L46 heat units

Changed.

• Highlight, page 10

L50 to coincide with predictions of

Added.

• Strike Out, page 10

L53 the

Removed.

• Highlight, page 10

L54 any reasons why phenology might be different in the midwestern US?

Presumably a result of climate, more detail provided in the Discussion.

• Highlight, page 11

L69 would be nice to report GPS points for sites, latitude info could prove useful to others

GPS location included in new Table 1.

• Highlight, page 11

L88 What resolution and time scale were the PRISM data?

Added.

• Highlight, page 11

L90 degree-days (DD) - what method used for DD?

Degree-day thresholds listed (L135-136).

• Highlight, page 13

L33 but wouldn't this be more in the SWD foraging environement rather than ambient? Were the orchards irrigated?

L133 removed. Yes, orchards were irrigated.

• Strike Out, page 14

L154 we?

Corrected.

• Highlight, page 16

L183 explanation for the confidence interval?

Added.

• Highlight, page 16

L186 explanation for CI missing

Added.

• Highlight, page 17

L206 smoothing protocol? Loess?

Smoothing protocol (penalized cubic regression) noted in the GAMM description in the Methods.

• Highlight, page 18

L244 but presumably if there was a later crop or host, the populations would likely increase well beyong the monitored period (Oct), or at least you don;t know from this data set, these data are specific to the tart cherry crop and should not be expanded to generalize the whole population

This is true. Included statement explaining we cannot estimate population trends expanding beyond our monitoring times.

• Highlight, page 21

L297 Careful, no data on crop infestation are presented. What if damage preceded trap captures? Unhappy growers.

Reference included backing statement (L367 – 369). 

• Highlight, page 21

L309 Would be important to link the model to 1) crop susceptibility (color), 2) infestation/damage.

Included these measurables L388.

Response to Reviewer #3’s comments

The paper seeks to develop a general model for SWD in the Midwest by fitting GLMM and GAMM statistical models to four years of field trap catch data using degree days, RH, year, and site as independent variables. The paper is well written and the statistical analyses well done.

The approach of putting field data on a dd scale is old (e.g., R.D. Hughes and N. Gilbert @1968), and normalizing the data for each location and year helps to put the phenology on a common scale based on the temperatures experienced at the different sites and years (L 274-276). The first question is what is the starting time for accumulating dd, and further how were the dd calculated (the term non-linear time gives direction but not precise information). Statistical models tend to be time and place specific

explaining why similar models from other locations do no work for the Midwest (lines 252-265). In fact, the proposed model(s) may not work in other areas of the Midwest. The parameter values should be given for the models.

Mechanistic models have been developed that use biodemographic functions to estimate the effects of physical factors (temperature, RH, photoperiod, etc.) at short time steps on the dynamics of the target poikilotherm. In an age -stage context, these models use abiotic and biotic factors as drivers

of the dynamics (doi:10.1016/j.ecolmodel.2016.05.014 and doi:10.1007/s10530-016-1255-6). No model should be used to forecast densities, hence the comment L 268-271 is inappropriate – at best one can

attempt to predict phenology “ for preemptive – IPM”. The authors recognize this limitation L-308-309.

We appreciate your comments. Degree-day accumulation began on January 1st for each year (L134), and information on the calculation of DD are provided in the temperature and humidity section in the Methods (L133 – 138). We understand no model based on regionally specific data can accurately predict phenology of D. suzukii in a different region, and this is especially true for an insect such as D. suzukii that has such a different phenology in different regions, hence the need for regional models. We believe these models can be applied the Midwest and to regions growing tart cherries experiencing similar climates to the Midwest to get a general sense of D. suzukii population trends throughout the field season. Parameters for the GAM referenced provided (L321 – 323). We agree with your comment on L268 – 271, the sentence was changed to be more appropriate (L334 – 337).

Some specific points:

L178- (t = 5.50, p < 0.001, are high relative to R2 = 0.04 – please check.

Thank you, corrected.

L201-203 This result is the critical result, and merits deeper explanation of the conditional mode.

Indeed, explanation provided L249 – 251 and L254 – 256.

The work seeks to develop IPM relevant information, but fail to provide accessible rules. Figures 1 and 2 were not particularly useful, showing a very wide scatter of data interpreted statistically in the text.

We prefer to keep Figures 1 and 2 because we feel that providing a visualization of the complete four year trap catch dataset is helpful for readers to go along with the trends explained statistically in the text.

The data reported in the supplemental materials is for 2018.

Thank you, data now includes all years.

*** Note from Editorial Office: please note that the responses to the following questions are 'partly' but this is not an option in the reviewer form and 'No' has therefore been selected:

2. Has the statistical analysis been performed appropriately and rigorously? Partly

3. Have the authors made all data underlying the findings in their manuscript fully available? Partly -2018 ***

---

## [Decision Letter · Decision Letter 1]

9 Apr 2020

Degree day models to forecast the seasonal phenology of Drosophila suzukii in tart cherry orchards in the Midwest U.S.

PONE-D-19-35638R1

Dear Dr. Guédot,

We are pleased to inform you that your manuscript has been judged scientifically suitable for publication and will be formally accepted for publication once it complies with all outstanding technical requirements.

With kind regards,

Antonio Biondi, Ph.D.

Academic Editor

PLOS ONE

Additional Editor Comments (optional):

Dear authors,

Your manuscript has been reviewed by one of the referees that had reviewed the original version. Such reviewer found the manuscript not worthy of publication. However, after reading the new comments, the (very carefully)revised manuscript and the rebuttal letter, I totally disagree with such recommendation and, as a consequence, I am promoting the acceptance of this version for publication in PONE.

Reviewers' comments:

Reviewer's Responses to Questions

**Comments to the Author**

1. If the authors have adequately addressed your comments raised in a previous round of review and you feel that this manuscript is now acceptable for publication, you may indicate that here to bypass the “Comments to the Author” section, enter your conflict of interest statement in the “Confidential to Editor” section, and submit your "Accept" recommendation.

Reviewer #1: All comments have been addressed

2. Is the manuscript technically sound, and do the data support the conclusions?

Reviewer #1: Partly

3. Has the statistical analysis been performed appropriately and rigorously? 

Reviewer #1: N/A

4. Have the authors made all data underlying the findings in their manuscript fully available?

Reviewer #1: No

5. Is the manuscript presented in an intelligible fashion and written in standard English?

Reviewer #1: Yes

6. Review Comments to the Author

Reviewer #1: The authors have fixed many of the comments that the reviewers had suggested to them. And the article has improved a lot, compared to the original version. However, at least for this reviewer, there are two key points that are important in interpreting this research, that is not fixed properly.

The authors have included an orchard with different cherry varieties in the study. Although they should be explained better the differences between these varieties (phenology, organoleptic, etc..). However, authors have justified by demonstrating that there is no difference between varieties of SWD catches during the study. At least for this reviewer, this could be justify 100% with statistics to show that there are no differences between cherry varieties on D. suzukii catch.

Finally, the second key point, is the most important of the article, because this manuscript is focused on the degree days models. At least for this reviewer, the weather stations are far from the fields where the traps were placed, and maybe this could be a reason to obtain strong confident models. However, this is a methodology mistake, that can't be fixed.

Other comments:

I also agree with the other reviewer about the insecticide treatments, which are necessary to consider to estimate the level of D. suzukii in relation to the treatments during the experiments.

Finally, no data on crop infestation are presented, which is crucial to know the damage on fruit in correlation with the DD models that authors did it.

7. PLOS authors have the option to publish the peer review history of their article (what does this mean?). If published, this will include your full peer review and any attached files.

Reviewer #1: No

---

## [Editor Report · Acceptance letter]

15 Apr 2020

PONE-D-19-35638R1 

Degree day models to forecast the seasonal phenology of *Drosophila suzukii *in tart cherry orchards in the Midwest U.S. 

Dear Dr. Guédot:

I am pleased to inform you that your manuscript has been deemed suitable for publication in PLOS ONE. Congratulations! Your manuscript is now with our production department. 

With kind regards,

on behalf of

Dr. Antonio Biondi 

Academic Editor

PLOS ONE